# Generative Emotion Cause Triplet Extraction in Conversations with Commonsense Knowledge

**Fanfan Wang, Jianfei Yu[*] and Rui Xia[*]**
School of Computer Science and Engineering,
Nanjing University of Science and Technology, China
{ffwang, jfyu, rxia}@njust.edu.cn

## Abstract

Emotion Cause Triplet Extraction in Conversations (ECTEC) aims to simultaneously extract emotion utterances, emotion categories, and cause utterances from conversations. However, existing studies mainly decompose the ECTEC task into multiple subtasks and solve them in a pipeline manner. Moreover, since conversations tend to contain many informal and implicit expressions, it often requires external knowledge and reasoning-based inference to accurately identify emotional and causal clues implicitly mentioned in the context, which are ignored by previous work. To address these limitations, in this paper, we propose a commonSense knowledge-enHanced generAtive fRameworK named SHARK, which formulates the ECTEC task as an index generation problem and generates the emotion-cause-category triplets in an end-to-end manner with a sequence-to-sequence model. Furthermore, we propose to incorporate both retrieved and generated commonsense knowledge into the generative model via a dual-view gate mechanism and a graph attention layer. Experimental results show that our SHARK model consistently outperforms several competitive systems on two benchmark datasets. Our source codes are publicly released at https://github.com/NUSTM/SHARK.

## 1 Introduction

Emotion understanding is a key component of human-like artificial intelligence, as emotions are intrinsic to humans and significantly influence our cognition, decision-making, and social interactions. Conversations, as a fundamental form of human communication, are replete with diverse emotions. Beyond mere emotion recognition, delving into the triggers behind these emotions in conversations is a more intricate and less explored task. A comprehensive understanding of both the speaker's emotions and their causes facilitates many applications

[*] Corresponding authors.

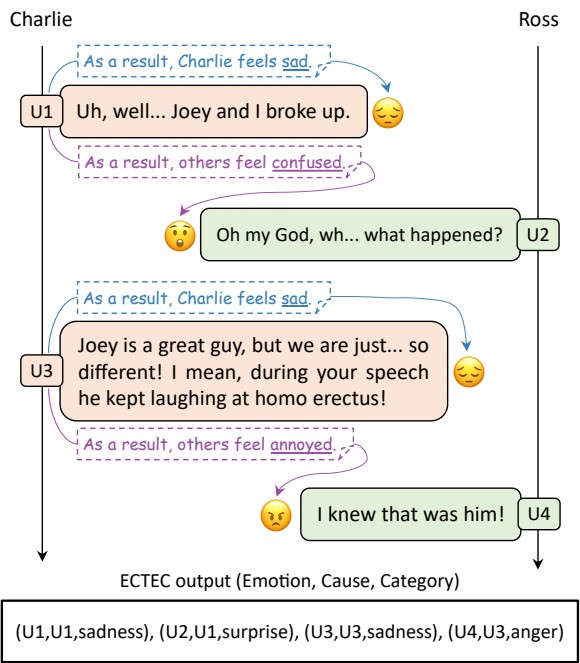

Figure 1: An example of extracting the emotion-cause-category triplets from the conversation with the help of commonsense knowledge in the dashed box.

such as customer support, mental health care, and human-computer interaction. Therefore, Emotion Cause Analysis in Conversations (ECAC) has been gaining increasing attention from both academia and industry in recent years.

Most existing studies on ECAC primarily focus on Causal Emotion Entailment (CEE) and Emotion-Cause Pair Extraction in Conversations (ECPEC). The former line of work on CEE assumes that the emotion utterances are given and formulates the ECAC task as an utterance classification problem (Poria et al., 2021; Li et al., 2022a; Zhang et al., 2022; Zhao et al., 2023; Gu et al., 2023), which aims to predict whether each utterance in a conversation is the cause of the given emotion utterance. The latter line of work on ECPEC focuses on designing different multi-task learning architectures to jointly extract the emotion utterances and

their corresponding causes in a pipeline manner (Li et al., 2022b; Jeong and Bak, 2023).

Due to the importance of emotion categories in ECAC, Wang et al. (2022) recently explored a task named Emotion Cause Triplet Extraction in Conversations (ECTEC), which aims to simultaneously extract emotion utterances, emotion categories, and cause utterances from conversations. For example, given the conversation in Figure 1, it is expected to identify four emotion-cause-category triplets. To tackle the task, they further proposed a two-step approach, which first extracts emotion utterances with emotion category and cause utterances separately, followed by pairing them to obtain valid emotion-cause-category triplets.

However, all the aforementioned studies on ECAC still suffer from two limitations. First, existing works on CEE and ECPEC primarily decompose ECTEC into several subtasks and only focus on tackling one or two subtasks. Although Wang et al. (2022) attempted to address the ECTEC task, their pipeline approach still suffers from the error propagation issue. To the best of our knowledge, there is still a lack of an end-to-end approach to generate all the emotion-cause-category triplets in one shot. Second, since interlocutors usually rely on the dialogue history and commonsense knowledge (CSK) to make sense of others' utterances and respond succinctly rather than explicitly (Grice, 1975), it often requires external knowledge and reasoning-based inference to accurately identify emotional and causal clues in the conversation. For example, in Figure 1, we can find that the CSK reasoned from U3 not only indicates Charlie's sadness in U3 and Ross's anger in U4, but also implies that U3 contains the causes behind their emotions. With the clues inferred from CSK, the two triplets "(U3,U3,surprise)" and "(U4,U3,anger)" can be easily inferred. Despite the importance of CSK, it is still under-explored how to utilize CSK to facilitate the ECTEC task.

To address these limitations, in this paper, we propose a commonSense knowledge-enHanced generAtive fRameworK named SHARK, which incorporates CSK into a pre-trained sequence-to-sequence model BART (Lewis et al., 2020) to generate all the emotion-cause-category triplets in an end-to-end manner. Specifically, SHARK formulates the ECTEC task as an index generation problem, which linearizes each emotion-cause-category triplet into a position index triplet. It then employs BART to encode the input conversation, followed by decoding a set of triplets containing the indexes of emotion utterances, cause utterances, and emotion categories. Moreover, to incorporate CSK into the BART-based framework, SHARK feeds each utterance to a pre-trained neural knowledge model COMET-ATOMIC$_{20}^{20}$ (Hwang et al., 2021) for commonsense knowledge generation and a widely-used knowledge base ATOMIC (Sap et al., 2019) for commonsense knowledge retrieval. Next, SHARK integrates both generated and retrieved knowledge via a dual-view gate mechanism to obtain the knowledge representation, and then introduces a knowledge-aware graph attention layer to capture the intra-speaker and inter-speaker interactions in the conversation. Finally, the knowledge-enhanced utterance representation is used to generate the position index sequence.

Our contributions are summarized as follows:

- We formulate the ECTEC task as an index generation problem by linearizing all emotion-cause-category triplets into a position index sequence, and employ a BART-based encoder-decoder framework to generate the index sequence from the input conversation.

- We further introduce a dual-view gate mechanism to integrate both retrieved and generated knowledge to obtain the knowledge representation, and then incorporate it into the BART-based framework via a knowledge-aware graph attention layer.

- Experimental results demonstrate that our generative framework consistently performs better than a number of competitive systems on two benchmark datasets. Further in-depth analysis shows the importance of the commonsense knowledge for the ECTEC task.

## 2 Methodology

### 2.1 Task Formulation and Model Overview

Given a conversation containing $n$ utterances $D = [U_1, \ldots, U_i, \ldots, U_N]$, in which each utterance $U_i$ corresponds to a speaker $S_i$, the goal of ECTEC is to extract all the emotion cause triplets:

$$\mathcal{P} = \left\{ \ldots, (U_j^e, U_k^c, y^e), \ldots \right\}, \qquad (1)$$

where $U_j^e$ is an emotion utterance with certain emotion $y^e$, $U_k^c$ is the corresponding cause utterance. The emotion category $y^e$ is one of the six basic

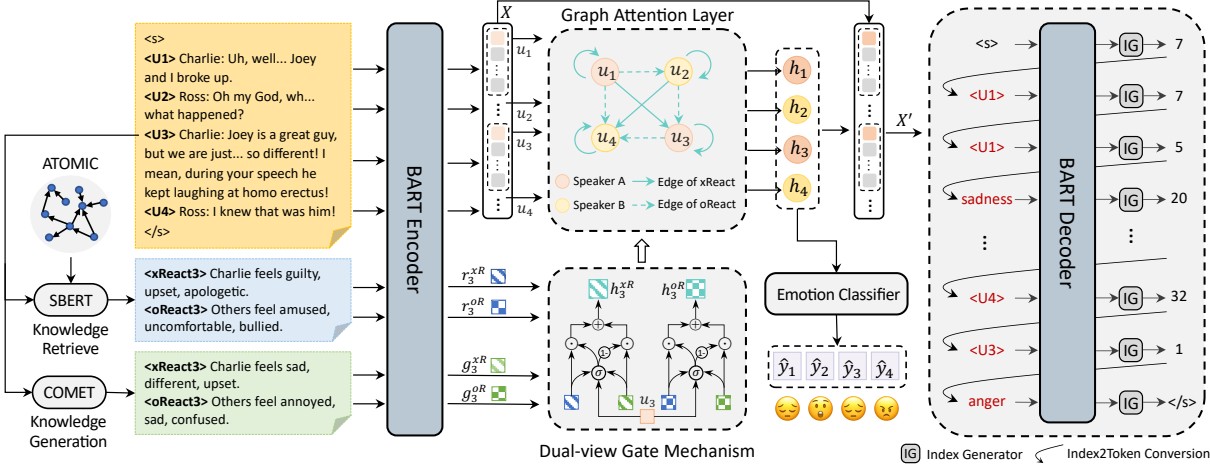

Figure 2: The overview of our proposed commonSense knowledge-enHanced generAtive fRameworK (SHARK).

emotions defined by Ekman et al. (1999), including *Anger*, *Disgust*, *Fear*, *Joy*, *Sadness* and *Surprise*.

As illustrated in Figure 2, our proposed common-Sense knowledge-enHanced generAtive fRameworK (SHARK) formulates the ECTEC task as an index generation problem, which linearizes all the emotion-cause-category triplets into their corresponding position indexes to obtain a position index sequence, and employs a pre-trained sequence-to-sequence model BART as the backbone to generate the position index sequence in an end-to-end manner. To incorporate CSK into the generative model, SHARK utilizes an external knowledge base and a pre-trained model to both retrieve and generate emotion-oriented CSK for each utterance, as shown in the left of Figure 2. Next, SHARK further introduces a dual-view gate mechanism to fuse the CSK representation, and then leverages it to enhance the utterance representation via a graph attention layer in the middle of Figure 2.

In the next subsections, we will first present the BART-based index generation framework, and then describe the details of the dual-view gate mechanism and the knowledge-aware graph attention layer.

## 2.2 Index Generation Framework

Given a conversation, we use the pre-trained BART model as the backbone to generate the output index sequence of the emotion-cause-category triplets.

**Encoder.** The encoder is a multi-layer bidirectional Transformer, which encodes the input sequence into the hidden representation. Specifically, we concatenate all the utterances in a conversation, and add several special tokens before each utterance to obtain the input sequence: $X$ = *<U1>*$S_1$ : $U_1$ ... *<Un>*$S_n$ : $U_n$**, where ** and ** refer to the start and end tokens, *<Ui>* is a special token to indicate the start of the $i$-th utterance, and $S_i$ and $U_i$ denote the speaker's name and the token sequence of the $i$-th utterance. The input sequence $X$ is then fed to the BART encoder to obtain the hidden representation:

$$\mathbf{X} = \text{Encoder}(X), \tag{2}$$

where $\mathbf{X} \in \mathbb{R}^{L \times d}$, $L$ is the length of $X$, and $d$ is the hidden dimension. The representation of the special token *<Ui>* in $\mathbf{X}$ is regarded as the utterance representation of $U_i$, i.e., $\boldsymbol{u}_i$. Based on this, we further incorporate CSK to obtain the knowledge-enhanced representation $\mathbf{X}'$ via a dual-view gate mechanism (Section 2.3) and a knowledge-aware graph attention layer (Section 2.4).

**Decoder.** To map all the emotion-cause-category triplets to a position index sequence, we use six indexes (i.e., 1 to 6) to denote six emotion categories, and then use $L$ indexes starting from 7 to denote each word in the input sequence $X$, in which the index of the special token *<Ui>* refers to the emotion or cause utterance. Formally, let us use $Y = [eu_1, cu_1, ec_1, \ldots, eu_m, cu_m, ec_m]$ to denote the output index sequence, where $eu$ and $cu$ represent the position indexes of the emotion and cause utterances, respectively, $ec$ denotes the index of the emotion category, and $m$ is the number of triplets. At the $t$-th time step, the decoder takes the knowledge-enhanced representation $\mathbf{X}'$ and the previous decoder outputs $Y_{<t}$ as inputs to predict the output probability distribution. Since $Y_{<t}$ is an index sequence, we first need to convert the indexes

into tokens as follows:

$$\hat{y}_t = \begin{cases} C_{y_t}, & y_t \leq |C|, \\ X_{y_t - |C|}, & y_t > |C|, \end{cases} \quad (3)$$

where $C$ is the token list of the emotion categories, $|C| = 6$.

Next, we can obtain the last hidden state of the BART decoder and predict the probability distribution as follows:

$$\boldsymbol{h}_t^d = \text{Decoder}(\mathbf{X}'; \hat{Y}_{<t}), \quad (4)$$

$$\bar{\mathbf{X}} = \text{MLP}(\mathbf{X}'), \quad (5)$$

$$\bar{\mathbf{H}}_{\text{utt}}^e = (\mathbf{E}_{\text{utt}}^e + \bar{\mathbf{X}}_{\text{utt}})/2, \quad (6)$$

$$P(y_t|X, Y_{<t}) = \text{Softmax}([\mathbf{C}^e; \bar{\mathbf{H}}_{\text{utt}}^e] \cdot \boldsymbol{h}_t^d), \quad (7)$$

where MLP refers to a multi-layer perceptron, $\mathbf{E}_{\text{utt}}^e$ = TokenEmbed$(X_{\text{utt}})$ and $\bar{\mathbf{X}}_{\text{utt}}$ respectively denote the embeddings of <$Ui$> and the hidden representations of <$Ui$>, $\mathbf{C}^e$ = TokenEmbed$(C)$ refers to the embeddings of six emotion categories; $[\cdot; \cdot]$ is the concatenation operation, and "$\cdot$" denotes the dot product; $P(y_t|X, Y_{<t}) \in \mathbb{R}^{(|C|+L)}$ is the final distribution on all indexes.

## 2.3 Dual-view Gate Mechanism

For each utterance, we separately obtain its retrieved and generated knowledge, and then integrate both of them through a gate mechanism.

**Knowledge Retrieval.** We utilize ATOMIC-2020[1] (Hwang et al., 2021), a widely-used commonsense knowledge graph covering social, physical, and eventive aspects of everyday inferential knowledge, as the external knowledge source. ATOMIC-2020 contains a large number of commonsense knowledge tuples composed of a head phrase, a relation type, and a tail phrase, e.g., ("*PersonX affords a car*", *xReact*, "*proud*"). In this paper, we focus on two social-interaction relations that are highly correlated with emotion and cause: *xReact* and *oReact*, which denote how the subject and the object feel after the input event occurs. Specifically, we use SBERT (Reimers and Gurevych, 2019) to calculate the similarity between each utterance in the dataset and all head phrases in ATOMIC-2020, and obtain the tail phrases under the two relations of the top-3 similar head phrases. We then concatenate these tail phrases as our retrieved knowledge for each utterance.

**Knowledge Generation.** A pre-trained neural knowledge model COMET-ATOMIC$_{20}^{20}$[2] is used to

generate novel commonsense knowledge tuples. Taking each utterance and the selected relation type as inputs, the model would automatically generate several tail phrases (we set the beam size to 3) as the CSK for the utterance. For example, given the utterance "*Uh, well... Joey and I broke up.*" and the relation type *xReact*, the following phrases {*sad, regretful, upset*} could be generated.

**Knowledge Encoding.** We first convert the knowledge phrases for each utterance into a complete sentence, which incorporates speaker's name, e.g., "[*Charlie*] *feels* [*sad, different, upset*]." for *xReact* and "*Others feel* [*annoyed, sad, confused*]." for *oReact*. Next, we concatenate the knowledge sentences of all utterances in the conversation and feed them to the BART encoder. The retrieved knowledge and generated knowledge under the two relation types are encoded separately in the same way as utterance encoding described in Section 2.2. Finally, for each utterance $U_i$, we obtain four knowledge representations: $\boldsymbol{r}_i^{xR}, \boldsymbol{r}_i^{oR}, \boldsymbol{g}_i^{xR}, \boldsymbol{g}_i^{oR}$.

**Knowledge Fusion.** Since the CSK under *xReact* and *oReact* types would independently influence the representation of utterances from the same speaker and other speakers, we employ a dual-view gate mechanism to perform knowledge fusion from two different views, thereby obtaining view-specific knowledge representations. Specifically, for each relation type, we calculate the knowledge weight through a linear layer, and then use the weight to integrate retrieved knowledge and generated knowledge. The formula is shown as follows:

$$\alpha_i^{xR} = \sigma(\mathbf{W}[\boldsymbol{u}_i, \boldsymbol{r}_i^{xR}, \boldsymbol{g}_i^{xR}]),$$
$$\alpha_i^{oR} = \sigma(\mathbf{W}[\boldsymbol{u}_i, \boldsymbol{r}_i^{oR}, \boldsymbol{g}_i^{oR}]), \quad (8)$$

$$\boldsymbol{h}_i^{xR} = \alpha_i^{xR}\boldsymbol{r}_i^{xR} + (1 - \alpha_i^{xR})\boldsymbol{g}_i^{xR},$$
$$\boldsymbol{h}_i^{oR} = \alpha_i^{oR}\boldsymbol{r}_i^{oR} + (1 - \alpha_i^{oR})\boldsymbol{g}_i^{oR}, \quad (9)$$

where $\mathbf{W} \in \mathbb{R}^{3d}$ is a trainable weight, $\sigma$ denotes the sigmoid function.

## 2.4 Knowledge-aware Graph Attention Layer

To better model the conversation context and incorporate CSK effectively, we employ a graph attention layer to capture the dynamics and dependencies among speakers in the conversation.

As shown in Figure 2, we construct a graph $\mathcal{G} = (\mathcal{V}, \mathcal{E})$ for each conversation, where $\mathcal{V}$ is the set of nodes and $\mathcal{E}$ is the set of edges between two nodes.

---

[1] https://allenai.org/data/atomic-2020
[2] https://github.com/allenai/comet-atomic-2020

- Nodes: Each utterance in the conversation is regarded as a node in the graph, and we initialize each node with the utterance representation $\boldsymbol{u}_i$ obtained from the encoder.

- Edges: The utterance nodes are linked in the temporal order of the conversation, and each edge is assigned an attention weight $e_{ij}$ $(i > j \geq 1)$ representing the relationship between two utterances. We introduce CSK under two relation types to enrich the semantic dependencies among utterances. Specifically, given an utterance node, for each of its neighbor nodes, if their speakers are the same, the knowledge representation $\boldsymbol{h}_i^{xR}$ is used to calculate the edge weight between them; otherwise, $\boldsymbol{h}_i^{oR}$ is utilized.

For a target node $\boldsymbol{u}_i$, the edge weight from neighbor nodes $\boldsymbol{u}_j$ can be computed as follows:

$$\boldsymbol{u}_j^{'} = \mathbf{W}^h \boldsymbol{h}_j, \tag{10}$$

$$\bar{\boldsymbol{h}}_j^{xR} = \boldsymbol{u}_j^{'} + \mathbf{W}^k \boldsymbol{h}_j^{xR}, \tag{11}$$

$$\bar{\boldsymbol{h}}_j^{oR} = \boldsymbol{u}_j^{'} + \mathbf{W}^k \boldsymbol{h}_j^{oR}, \tag{12}$$

$$e_{ij}^{xR} = \text{LeakyReLU}(\boldsymbol{a}_1^T[\mathbf{W}^h \boldsymbol{u}_i; \bar{\boldsymbol{h}}_j^{xR}]), \tag{13}$$

$$e_{ij}^{oR} = \text{LeakyReLU}(\boldsymbol{a}_2^T[\mathbf{W}^h \boldsymbol{u}_i; \bar{\boldsymbol{h}}_j^{oR}]), \tag{14}$$

$$e_{ij} = \begin{cases} e_{ij}^{xR}, & S_i = S_j, \\ e_{ij}^{oR}, & S_i \neq S_j, \end{cases} \tag{15}$$

where $\boldsymbol{a}_1, \boldsymbol{a}_2 \in \mathbb{R}^{2d}$ and $\mathbf{W}^h, \mathbf{W}^k \in \mathbb{R}^{d \times d}$ are learnable parameters.

Next, we normalize the edge weights across all choices of $j$ using the Softmax function. Finally, the knowledge-enhanced utterance representation $\boldsymbol{h}_i$ is obtained by aggregating neighbor nodes according to the normalized edge weights:

$$\alpha_{ij} = \text{Softmax}_j(e_{ij}), \tag{16}$$

$$\boldsymbol{h}_i = \sum_{j \in \mathcal{N}_i} \alpha_{ij} \boldsymbol{u}_j^{'}, \tag{17}$$

where $\mathcal{N}_i$ is the set of neighbor nodes of utterance node $\boldsymbol{u}_i$ in the graph.

To facilitate our model to capture emotional dynamics, we add an auxiliary task of emotion recognition in conversations (ERC). As shown in Figure 2, an emotion classifier is applied to predict the emotion category of each utterance based on the knowledge-enhanced utterance representation.

$$P(y_i^{\text{emo}}) = \text{Softmax}(\text{MLP}(\boldsymbol{h}_i)), \tag{18}$$

where $P(y_i^{\text{emo}}) \in \mathbb{R}^7$ is the probability distribution over all emotion categories including *Neutral*.

| Number of items | ECF | RECCON |
|---|---|---|
| Conversations | 1,374 | 1,106 |
| Utterances | 13,619 | 11,104 |
| Emotion (utterances) | 7,690 | 5,861 |
| Emotion-cause (utterance) pairs | 9,794 | 9,915 |

Table 1: Statistics of two benchmark datasets.

## 2.5 Model Training

During the training phase, we use the teacher forcing strategy to train our model and the negative log-likelihood loss to optimize the model. The loss function is defined as follows:

$$\mathcal{L} = \mathcal{L}^{\text{gen}} + \mathcal{L}^{\text{aux}}, \tag{19}$$

$$\mathcal{L}^{\text{gen}} = -\frac{1}{M} \sum_{t=1}^{M} \log P(y_t^* | X, Y_{<t}), \tag{20}$$

$$\mathcal{L}^{\text{aux}} = -\frac{1}{N} \sum_{i=1}^{N} \log P(y_i^{\text{emo}*}), \tag{21}$$

where $M$ and $N$ refer to the length of the output index sequence and the number of utterances in a conversation, respectively; $y_t^*$ and $y_i^{\text{emo}*}$ denote the ground truth label of index and emotion, respectively. Moreover, during the inference, we use the beam search to obtain the target index sequence in an autoregressive manner.

## 3 Experiments

### 3.1 Experimental Settings

**Datasets.** We conduct experiments on two benchmark datasets. **ECF** (Wang et al., 2022) is a multimodal conversational emotion cause dataset containing multi-party conversations from the sitcom *Friends*, which is closer to real-world scenarios. In this paper, we only consider the textual input. **RECCON** (Poria et al., 2021) includes dyadic conversations and is built for the task of emotion cause recognition in conversations. We use the subset RECCON-DD derived from DailyDialog (Li et al., 2017). Both datasets are divided into training, validation and test sets. The basic statistics of the two datasets are shown in Table 1.

**Implementation Details.** We utilize the pre-trained BART-base[3] model to initial the parameters in the index generation framework. During training, we use the Adam optimizer with linear warm up and a weight decay of 1e-2 for parameter tuning. The batch size and initial learning rate are set to 16 and 2e-5, respectively. We use beam search to

---

[3] https://huggingface.co/facebook/bart-base

| Dataset | | Method | Anger | Disgust | Fear | Joy | Sadness | Surprise | 6 Avg. | 4 Avg. |
|---|---|---|---|---|---|---|---|---|---|---|
| ECF | Pipeline | ECPE-2steps (Wang et al., 2022) | 24.39 | 0.00 | 0.71 | 38.84 | 21.60 | 40.24 | 29.32 | 31.92 |
| | | ECPE-2steps + CSK | 23.86 | 0.00 | 3.71 | 38.41 | 23.13 | 40.71 | 29.50 | 32.03 |
| | E2E | ECPE-2D (Ding et al., 2020a) | 25.13 | 0.00 | 0.00 | 41.25 | 21.62 | 43.24 | 30.80 | 33.55 |
| | | ECPE-2D + CSK | 25.36 | 0.00 | 0.00 | 38.89 | 22.62 | 43.26 | 30.37 | 33.09 |
| | | UECA-Prompt (Zheng et al., 2022) | 27.37 | 12.85 | 7.91 | 37.96 | 22.51 | 39.53 | 30.75 | 32.49 |
| | | UECA-Prompt + CSK | 23.57 | 7.70 | 9.36 | 34.95 | 22.44 | 38.69 | 28.49 | 30.30 |
| | | BART | 27.15 | 4.37 | 2.20 | 38.66 | 25.51 | 37.53 | 30.35 | 32.74 |
| | | SHARK (Ours) | 28.65 | 10.42 | 5.33 | 40.41 | 25.35 | 40.45 | **32.24** | **34.33** |
| RECCON | Pipeline | ECPE-2steps (Wang et al., 2022) | 17.19 | 1.33 | 0.00 | 45.81 | 17.54 | 25.01 | 34.25 | 36.08 |
| | | ECPE-2steps + CSK | 19.62 | 0.61 | 0.95 | 45.34 | 20.34 | 25.80 | 34.71 | 36.57 |
| | E2E | ECPE-2D (Ding et al., 2020a) | 22.58 | 0.00 | 0.00 | 47.72 | 12.82 | 36.10 | 36.58 | 38.59 |
| | | ECPE-2D + CSK | 21.51 | 0.35 | 0.00 | 47.82 | 13.78 | 32.95 | 36.32 | 38.30 |
| | | UECA-Prompt (Zheng et al., 2022) | 24.11 | 11.38 | 31.20 | 47.53 | 21.40 | 35.09 | 38.63 | 39.63 |
| | | UECA-Prompt + CSK | 22.28 | 6.98 | 27.43 | 45.14 | 23.42 | 35.74 | 36.92 | 38.06 |
| | | BART | 27.68 | 11.99 | 37.99 | 44.04 | 26.63 | 33.42 | 37.73 | 38.46 |
| | | SHARK (Ours) | 27.00 | 12.88 | 42.65 | 46.88 | 31.09 | 32.18 | **39.82** | **40.53** |

Table 2: Performance comparison of different methods on the ECTEC task. The best results are in bold.

generate the index sequence and set the beam size to 4. The results on the test set come from the best checkpoint in the validation set. We repeat all the experiments five times with different seeds and report the average results. All experiments are conducted on an Nvidia RTX-3090 GPU.

**Evaluation Metrics.** Similar to (Wang et al., 2022), we separately evaluate the emotion-cause pairs of each emotion category in the triplets with $F_1$ score and further calculate a weighted average of $F_1$ scores across different emotion categories. Considering the imbalance of emotion categories in the two datasets, we also report the weighted average $F_1$ score of the four main emotion categories except *Disgust* and *Fear*.

## 3.2 Compared Methods

Since less work has been done on the ECPEC or ECTEC tasks, we compare our framework with several representative methods for ECPE in news articles: (1) *ECPE-2steps* (Xia and Ding, 2019) is the first pipeline framework proposed for ECPE, which individually extracts the emotion set and cause set, followed by emotion-cause pairing and filtering. Wang et al. (2022) has adapted it to the ECTEC task. (2) *ECPE-2D* (Ding et al., 2020a) is a joint end-to-end (E2E) framework using the cross-road 2D transformers to model the interactions of different emotion-cause pairs. (3) *UECA-Prompt* (Zheng et al., 2022) is a universal prompt method that decomposes ECPE into multiple objectives and converts them into sub-prompts. (4) *BART*, which only utilizes the index generation framework to generate the triplets, without two knowledge-enhanced modules and the auxiliary task in our *SHARK* model.

| Method | ECF | | RECCON | |
|---|---|---|---|---|
| | Emo. $F_1$ | Cau. $F_1$ | Emo. $F_1$ | Cau. $F_1$ |
| ECPE-2steps | 55.46 | 62.83 | 64.34 | 58.92 |
| ECPE-2steps + CSK | 55.61 | 65.05 | 64.96 | 58.83 |
| ECPE-2D | 56.98 | 67.53 | 66.53 | 62.11 |
| ECPE-2D + CSK | 56.28 | 66.51 | 66.24 | 62.22 |
| UECA-Prompt | 55.83 | 67.23 | 68.46 | 63.62 |
| UECA-Prompt + CSK | 53.92 | 62.08 | 65.97 | 62.38 |
| BART | 58.63 | **70.02** | 67.96 | **69.01** |
| SHARK (Ours) | **60.74** | 69.13 | **71.94** | 67.72 |

Table 3: Results of the emotion extraction and cause extraction subtasks based on the predicted triplets.

It should be noted that *ECPE-2D* and *UECA-Prompt* are designed for ECPE and require modifications to extend them to the ECTEC task. Specifically, we adapt their emotion recognition module from binary classification to multi-class classification. Moreover, we have explored simply incorporating CSK into these methods, which is denoted as "· + *CSK*" in the tables of experimental results. For *ECPE-2steps* and *ECPE-2D*, we feed the generated knowledge to their encoder to obtain the knowledge representation for each utterance, followed by concatenating the knowledge representation and the original utterance feature as the final utterance representation. For *UECA-Prompt*, we place the commonsense knowledge after each utterance, and the whole conversation is then fed into the model together. All the models are initialized with parameters from the pre-trained BERT-base[4] or BART-base for a fair comparison.

## 3.3 Main Results

In Table 2, we report the results of different methods on the ECTEC task. To better compare these methods, we also report the $F_1$ scores of two sub-

---

[4] https://github.com/google-research/bert or https://huggingface.co/bert-base-cased

| Method | ECF | | | | | | | | RECCON | | | | | | | |
|---|---|---|---|---|---|---|---|---|---|---|---|---|---|---|---|---|
| | Ang. | Dis. | Fear | Joy | Sad. | Sur. | 6 Avg. | 4 Avg. | Ang. | Dis. | Fear | Joy | Sad. | Sur. | 6 Avg. | 4 Avg. |
| SHARK | 28.65 | 10.42 | 5.33 | 40.41 | 25.35 | 40.45 | **32.24** | **34.33** | 27.00 | 12.88 | 42.65 | 46.88 | 31.09 | 32.18 | **39.82** | **40.53** |
| - w/o GAL | 27.77 | 7.65 | 5.37 | 38.97 | 25.50 | 39.35 | 31.25 | 33.42 | 25.51 | 16.23 | 40.76 | 46.52 | 30.17 | 35.12 | 39.54 | 40.19 |
| - w/o DGM | 29.24 | 10.01 | 8.41 | 38.37 | 26.73 | 40.00 | 32.02 | 34.03 | 25.11 | 15.66 | 35.66 | 47.01 | 30.72 | 34.64 | 39.68 | 40.45 |
| - w/o retrieved CSK | 28.85 | 6.26 | 4.65 | 40.66 | 24.08 | 38.16 | 31.40 | 33.70 | 26.50 | 15.59 | 37.84 | 46.83 | 30.53 | 32.67 | 39.67 | 40.39 |
| - w/o generated CSK | 27.45 | 7.31 | 8.88 | 39.34 | 26.09 | 37.48 | 31.05 | 33.13 | 24.89 | 15.94 | 40.54 | 46.63 | 31.33 | 31.42 | 39.33 | 39.97 |
| - w/o CSK | 26.76 | 10.90 | 5.52 | 37.70 | 26.71 | 39.84 | 31.12 | 33.09 | 22.32 | 13.58 | 43.75 | 47.08 | 29.20 | 35.82 | 39.30 | 39.94 |
| - w/o auxiliary task | 28.74 | 7.66 | 4.00 | 38.77 | 26.65 | 35.33 | 30.75 | 32.92 | 28.03 | 12.51 | 33.97 | 45.34 | 27.22 | 32.68 | 38.52 | 39.35 |

Table 4: Ablation study on two datasets. "GAL" and "DGM" denotes the knowledge-aware graph attention layer and the dual-view gate mechanism, respectively.

tasks including emotion extraction and cause extraction in Table 3. Note that the weighted $F_1$ for emotion extraction is evaluated based on the emotion categories in the predicted triplets, rather than the predictions from the auxiliary task.

**Results on ECTEC.** First, it is clear that the E2E methods significantly outperform *ECPE-2steps*, which shows the superiority of the joint E2E framework in addressing the ECTEC task compared to the two-step pipeline framework. Second, *BART* achieves comparable performance to other baseline methods, indicating the feasibility of the index generation framework to solve ECTEC. Third, the introduction of CSK on the baseline methods does not lead to significant improvements, and *ECPE-2D + CSK*, *UECA-Prompt + CSK* even perform worse on the two datasets. This suggests that simply concatenating knowledge may bring much noise, and it is necessary to explore more effective approaches for knowledge fusion. Finally, we can clearly observe that our proposed *SHARK* obtains the best results and performs much better than *BART*, which demonstrates the effectiveness of our framework and reveals the benefits of incorporating CSK into the ECTEC task.

**Results on two subtasks.** Similar to the conclusions drawn on ECTEC, incorporating knowledge into the baseline methods through simple concatenation may lead to a performance decline. However, *SHARK* greatly outperforms other methods on both subtasks, further validating the effectiveness of our framework and the necessity of knowledge fusion. Furthermore, given that the relation types of CSK we selected (*xReact* and *oReact*) are directly related to emotions, *SHARK* shows significant improvement over *BART* primarily on the emotion extraction subtask.

### 3.4 Ablation Study

To investigate the impact of individual modules on the overall performance, we conduct an ablation study of *SHARK* for the ECTEC task, and the

| Method | ECF → RECCON | | RECCON → ECF | |
|---|---|---|---|---|
| | 6 Avg. | 4 Avg. | 6 Avg. | 4 Avg. |
| ECPE-2steps | 25.95 | 27.35 | 16.03 | 17.46 |
| ECPE-2steps + CSK | 27.12 | 28.54 | 17.19 | 18.68 |
| ECPE-2D | 28.82 | 30.38 | 16.36 | 17.82 |
| ECPE-2D + CSK | 28.71 | 30.28 | 15.52 | 16.85 |
| UECA-Prompt | 28.87 | 29.77 | 17.60 | 18.74 |
| UECA-Prompt + CSK | 27.92 | 28.83 | 15.69 | 16.28 |
| BART | 35.91 | 37.12 | 18.04 | 19.26 |
| SHARK (Ours) | **36.56** | **37.68** | **18.70** | **19.87** |

Table 5: Cross-dataset evaluation for ECTEC. "A → B" refers to training on dataset A and testing on dataset B.

experimental results are shown in Table 4.

We can see that removing different modules from *SHARK* leads to varying degrees of performance degradation. Specifically, discarding the auxiliary task has the largest impact on performance, indicating that ERC is helpful for triplet extraction. Moreover, removing the generated CSK has a greater negative effect compared to removing the retrieved CSK. This suggests the ability of the pre-trained knowledge model to generate relevant and informative commonsense knowledge, which is beneficial to emotion-cause understanding. Furthermore, the significant performance drop when both sources of external knowledge are disregarded highlights the importance of incorporating knowledge into the ECTEC task.

### 3.5 In-Depth Analysis

**Cross-dataset Analysis.** To compare the generalization of each method, we conduct cross-dataset testing between ECF and RECCON, and report the results in Table 5.

An obvious observation is that the performance of all the methods significantly deteriorates when tested on a different dataset. This indicates that training a model on one dataset does not ensure good adaptation to other datasets. However, in comparison with other methods, *BART* and *SHARK* show great advantages across different datasets, especially *SHARK* outperforms others by at least 7% on 4 avg. $F_1$ under the *ECF→RECCON* setting, which demonstrates the strong generalization abil-

| Method | num_utt ≤ 10 | | num_utt > 10 | |
|---|---|---|---|---|
| | 6 Avg. | 4 Avg. | 6 Avg. | 4 Avg. |
| ECPE-2D | 33.56 | **37.41** | 29.41 | 31.71 |
| ECPE-2D + CSK | 32.84 | 36.61 | 29.10 | 31.37 |
| BART | 31.45 | 34.50 | 29.91 | 32.03 |
| SHARK (Ours) | **33.68** | 36.16 | **31.49** | **33.46** |

Table 6: Performance comparison of different methods for conversations with varying numbers of utterances from the test set of ECF.

| Method | Ang. | Dis. | Fear | Joy | Sad. | Sur. | 6 Avg. | 4 Avg. |
|---|---|---|---|---|---|---|---|---|
| ChatGPT (0-shot) | 13.19 | 21.62 | 28.57 | 24.14 | 14.29 | 16.95 | 18.32 | 17.89 |
| ChatGPT (5-shot) | 14.81 | 20.00 | 0.00 | 33.33 | 26.92 | 15.69 | 22.18 | 22.76 |
| SHARK (Ours) | 18.72 | 5.06 | 0.00 | 39.03 | 12.89 | 37.36 | **27.19** | **29.23** |

Table 7: Performance comparison of ChatGPT and our framework on 50 conversations from the test set of ECF.

ity and high robustness of our framework. Moreover, we find that models trained on ECF can adapt to RECCON well, while models trained on REC-CON perform poorly on ECF. We conjecture the reason is that the conversations in ECF come from TV series that are close to the real world, which involve informal text and complex scenes; while RECCON consists of human-generated conversations that are simpler and more formal, thus weakening the generalization of the model.

**The Impact of Conversation Length.** We conduct further analysis to explore the impact of different conversation lengths, i.e., separately evaluating the predictions for conversations with varying numbers of utterances from the test set of ECF. As shown in Table 6, we can observe that our generative models perform much better than encoder-based methods on conversations with more than 10 utterances, which account for about 42.65% in the ECF dataset. In these long conversations, encoder-based methods often fail to fully consider the contextual information and ignore a number of triplets, while our generative models tend to comprehensively capture the context, effectively capturing a broader range of triplets.

**Comparison with ChatGPT.** Considering the remarkable performance of large language models in various NLP tasks, we further apply ChatGPT to the ECTEC task under zero-shot and few-shot settings. Specifically, we randomly selected 50 conversations from the test set of ECF, and then fed each test conversation and a task-instruction prompt into ChatGPT to obtain the predicted emotion-cause triplets. Table 7 presents the 0-shot and 5-shot results of ChatGPT. It is obvious that SHARK performs significantly better than ChatGPT in both zero-shot and few-shot settings. We conjecture that

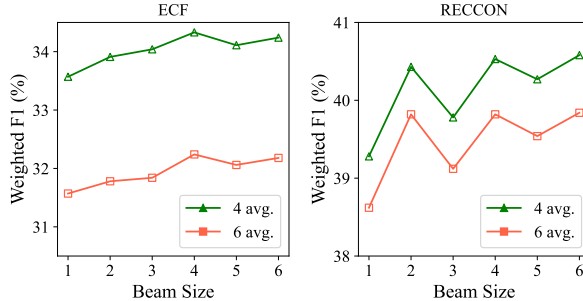

Figure 3: The F₁ change curve with different beam size.

due to the complexity of extracting three elements (i.e., emotion, cause, and category) in the ECTEC task, the performance of ChatGPT is not satisfactory.

**Sensitivity Analysis of Beam Size.** In order to investigate the impact of beam size on our SHARK model, we conduct experiments with different beam sizes, and the results are shown in Figure 3. The $F_1$ curves exhibit slightly different trends on the two datasets, but a beam size of 4 is the optimal choice for both datasets.

### 3.6 Case Study

To show the advantage of our SHARK model, we compare its predictions with the output of three baseline systems on a test sample. As shown in Figure 4, *ECPE-2D* and *ECPE-2D + CSK* only correctly identify two emotion-cause-category triplets, i.e., "(U2,U1,surprise)" and "(U2,U2,surprise)". Moreover, our base model *BART* can further identify another triplet "(U1,U1,surprise)", but still ignores "(U4,U1,surprise)" due to the long-term dependency between U4 and U1. In contrast, with the help of the CSK, *SHARK* correctly extracts all the four triplets. These observations show the effectiveness of the proposed generative model and CSK for the ECTEC task.

## 4 Related Work

### 4.1 Emotion Cause Analysis

Emotion Cause Analysis (ECA) has attracted increasing attention in recent years. It contains two representative subtasks: emotion cause extraction (ECE) and emotion-cause pair extraction (ECPE). ECE aims to extract the potential causes given the emotions (Lee et al., 2010a,b; Gui et al., 2016a,b, 2017; Fan et al., 2019); ECPE was proposed to jointly extract the emotions and the corresponding causes in pairs, thereby solving the problem of ECE's emotion annotation dependency (Xia and

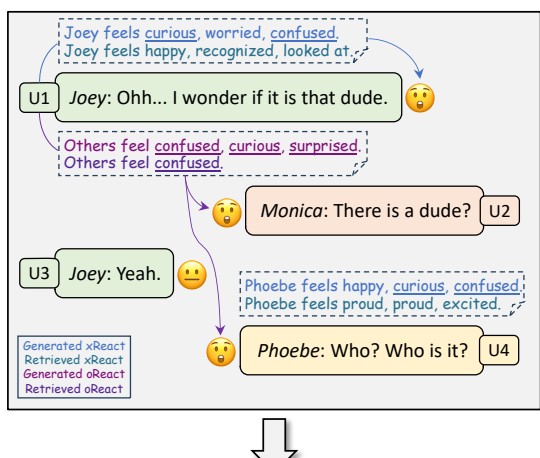

| Ground Truth | (U1,U1,surprise), (U2,U1,surprise), (U2,U2,surprise), (U4,U1,surprise) |
|---|---|
| ECPE-2D | (U1,U1,sadness), (U2,U1,surprise), (U2,U2,surprise) |
| ECPE-2D + CSK | (U2,U1,surprise), (U2,U2,surprise) |
| BART | (U1,U1,surprise), (U2,U1,surprise), (U2,U2,surprise), (U3,U2,surprise) |
| SHARK | (U1,U1,surprise), (U2,U1,surprise), (U2,U2,surprise), (U4,U1,surprise) |

Figure 4: Comparison of predictions on a test sample.

Ding, 2019). Much research has been conducted on the ECPE task. Xia and Ding (2019) first proposed a two-step framework ECPE-2steps, which first extracts an individual emotion set and cause set, and then pairs the corresponding emotions and causes. Following their work, many end-to-end approaches have been proposed to address the limitations of the pipeline architecture. One line of work focuses on multi-task learning using a joint modeling framework (Ding et al., 2020a,b; Wei et al., 2020; Fan et al., 2020; Chen et al., 2022). Another line of work transforms ECPE into a unified sequence labeling problem and designs novel tagging schemes (Yuan et al., 2020; Chen et al., 2020; Cheng et al., 2021; Fan et al., 2021). More recently, several studies attempted to address the ECPE task with prompt-based methods (Zheng et al., 2022) and machine reading comprehension-based methods (Zhou et al., 2022; Cheng et al., 2023).

### 4.2 Emotion Analysis in Conversations

Emotion recognition in conversations (ERC) is a hot-spot task in sentiment analysis, which aims to assign emotion labels to all the utterances in a conversation. Due to the increasing amount of public conversational data (Busso et al., 2008; Li et al., 2017; Poria et al., 2019; Firdaus et al., 2020), ERC has received continuous attention in the field of af-

fective computing (Majumder et al., 2019; Ghosal et al., 2020; Zhu et al., 2021; Song et al., 2022).

Recent years have witnessed a shift from ERC to ECAC. Poria et al. (2021) introduced an interesting task of recognizing emotion cause in conversations, aiming to find the causes behind the given emotions in the conversation, and constructed a new dataset RECCON. Several works for the CEE subtask have subsequently emerged (Li et al., 2022a; Zhang et al., 2022; Zhao et al., 2023; Gu et al., 2023). Furthermore, Li et al. (2022b) and Jeong and Bak (2023) attempted to extract emotion and causes in conversations simultaneously, and Wang et al. (2022) introduced the ECTEC task. However, the aforementioned encoder-only models for ECPEC mainly solve the task in a pipeline manner (independently predicting emotions and causes before matching, or first predicting emotions and then using the emotion predictions to infer causes), which suffer from error propagation. In contrast, our proposed generative framework enables end-to-end triplet generation, which can extract all the emotion-cause-category triplets from a conversation in one shot.

## 5    Conclusion

In this paper, we proposed a commonSense knowledge-enHanced generAtive fRameworK named SHARK for the Emotion Cause Triplet Extraction in Conversations (ECTEC) task. Specifically, we formulated the ECTEC task as an index generation problem and employed a BART-based model to generate all the emotion-cause-category triplets in one shot. Moreover, we designed a dual-view gate mechanism and a graph attention layer to incorporate both the retrieved and generated commonsense knowledge. Experimental results on two benchmark datasets show the superiority of SHARK over a number of comparison systems and the usefulness of commonsense knowledge.

## Limitations

Although the proposed SHARK model has obtained state-of-the-art performance on two benchmark datasets for the ECTEC task, our work still suffers from the following limitations.

First, we only consider the commonsense knowledge under two relation types, i.e., *xReact* and *oReact*, and thus design a dual-view gate mechanism to better capture the intra-speaker and inter-speaker interactions. However, there are other causal rela-

tions that may help cause inference, such as *xWant* and *oWant*. We plan to incorporate the other related causal relations into our generative framework in the future. Second, it might be interesting to explore the potential of incorporating commonsense knowledge for different modalities to boost the performance of the Multimodal ECTEC task.

## Ethics Statement

This paper does not involve any data collection and release, and thus there are no privacy issues. All the experiments are conducted on two publicly available datasets, namely ECF (Wang et al., 2022) and RECCON (Poria et al., 2021), which do not include personal information or contain any objectionable content that could potentially harm individuals or communities.

## Acknowledgements

The authors would like to thank the anonymous reviewers for their insightful comments. This work was supported by the Natural Science Foundation of China (62076133 and 62006117), and the Natural Science Foundation of Jiangsu Province for Young Scholars (BK20200463) and Distinguished Young Scholars (BK20200018).

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
