# OpenReview forum: "Generative Emotion Cause Triplet Extraction in Conversations with Commonsense Knowledge"
_EMNLP/2023/Conference — EMNLP 2023 Findings_

### Official Review · Reviewer_yf8h · 2023-08-02

**Soundness:** 3

**Excitement:**

3: Ambivalent: It has merits (e.g., it reports state-of-the-art results, the idea is nice), but there are key weaknesses (e.g., it describes incremental work), and it can significantly benefit from another round of revision. However, I won't object to accepting it if my co-reviewers champion it.

**Paper Topic And Main Contributions:**

This paper studies the problem of extracting triples of {emotion cause, emotion utterance, emotion category} from conversations. The main technical contribution is an end-to-end generation-based model with commonsense knowledge.

**Reasons To Accept:**

- the paper is in general well-written and easy to follow (though some notations are dense)
- the empirical results are good.

**Reasons To Reject:**

- the rationale behind generating rather than discriminating triple entries needs more discussions and empirical evaluations: roughly, encoder-only models (with some additional classification layers) could also be applied to perform the extraction. What are the advantages of the generation paradigm?
- can we remove the index2token step in Figure 2 (with supervised decoding, it is possible for BART to adapt to the new input sequence).
- it should be helpful to elaborate more details on the implementation of baselines and justify that they are properly tuned and fairly compared.
- it would be better to compare number of parameters (model size) and inference speed.


**Reproducibility:**

4: Could mostly reproduce the results, but there may be some variation because of sample variance or minor variations in their interpretation of the protocol or method.

**Reviewer Confidence:**

4: Quite sure. I tried to check the important points carefully. It's unlikely, though conceivable, that I missed something that should affect my ratings.

---

> ### Author Rebuttal · Authors · 2023-08-29
>
> Thanks for your careful review and valuable suggestions. We will explain your concerns point by point below.
>
> **Q1: What are the advantages of the generation paradigm?**
>
> **A1:** We first clarify our motivation behind the proposed generative model for the ECTEC task.
> To address the ECTEC task, an intuitive idea is to first identify which utterance expresses the emotion, and then find the cause utterance that triggers it, followed by determining the emotion category based on the emotion-cause utterance pair. Therefore, emotion is the core element of the ECTEC task, and the other two elements (i.e., cause and category) are highly dependent on the emotion utterance. We believe the decoder in the generative model can well capture the conditional dependency relationship among the three elements, because it generates the emotion utterance, cause utterance, and emotion category in an autoregressive manner to form the triplet.
>
> Furthermore, previous encoder-only models for ECPEC mainly solve the task in a pipeline manner (independently predicting emotions and causes before matching, or first predicting emotions and then using the emotion predictions to infer causes), which suffer from error propagation. Although a number of joint models have been proposed to boost the performance of the ECPE task, they are primarily based on multi-task learning, which still suffers from error propagation. In contrast, our proposed generative framework enables end-to-end triplet generation, which can extract all the emotion-cause-category triplets from a conversation in one shot.
>
> Moreover, as shown in the Cross-dataset Evaluation in Table 5, the generative framework demonstrates stronger generalization, suggesting extensive potential for real applications.
>
>
> **Q2: Can we remove the index2token step in Figure 2?**
>
> **A2:** No, it is necessary to keep the index2token step in our current framework. This is because if we remove the index2token step, the input of the decoder would be a sequence of position indexes rather than the tokens from the source text, in which the semantic meanings of the numerical indexes are ambiguous, preventing effective attention interaction with the output representations of the encoder.
>
>
> **Q3: More details on the implementation of baselines and justify that they are properly tuned and fairly compared.**
>
> **A3:** Thanks for your suggestion. We will add more details about the implementation of baselines. Specifically, all models are initialized with parameters from a pre-trained model of 'base' size (BERT-base/BART-base). For a fair comparison, we use the same batch size for all the models and conduct all the experiments on an Nvidia RTX-3090 GPU. The average results of multiple experiments are reported for each model. Since ECPE-2D and UECA-Prompt are designed for ECPE, we adapt their emotion recognition module from binary classification to multi-class classification to perform our task.
>
>
> **Q4: It would be better to compare number of parameters (model size) and inference speed.**
>
> **A4:** We have conducted experiments on an Nvidia RTX-3090 GPU to compare the computational efficiency of different methods and report the training and inference time as well as the parameter size.  As shown in the table below, we make the following observations:
>
> - SHARK incurs slightly higher computational cost than the baseline model BART, due to introducing more parameters for the knowledge extraction and fusion modules.
> - Although there are more parameters, the training time and inference time of SHARK are comparable to those of UECA-Prompt.
>
> Based on these observations, we believe that the computational cost of SHARK is generally acceptable considering the performance improvement over other baselines.
>
> | Method             | Number of Parameters (M) | Training Time (s) | Inference Time (s) |
> | ------------------ | ------------------------ | ----------------- | ------------------ |
> | UECA-Prompt        | 108.34                   | 69.19             | 139.52             |
> | UECA-Prompt  + CSK | 108.34                   | 70.31             | 140.67             |
> | BART               | 140.69                   | 55.40             | 118.06             |
> | SHARK (Ours)       | 152.14                   | 77.67             | 144.53             |
>
> *Note*：`Training Time` refers to the duration for the model to run through the training set and the verification set in a single training epoch, and `Inference Time` refers to the time it takes to make predictions on the entire test set. The training/inference time of UECA-Prompt is measured based on its publicly available source codes.

---

### Official Review · Reviewer_xHNc · 2023-08-03

**Soundness:** 4

**Excitement:**

3: Ambivalent: It has merits (e.g., it reports state-of-the-art results, the idea is nice), but there are key weaknesses (e.g., it describes incremental work), and it can significantly benefit from another round of revision. However, I won't object to accepting it if my co-reviewers champion it.

**Paper Topic And Main Contributions:**

This model tackles the task of Emotion-Cause Triplet Extraction in Conversations (ECTEC).
This paper proposes a commonsense knowledge-enhanced sequence-to-sequence model.
Experiment results show that the proposed model can achieve new state-of-the-art performance.

**Questions For The Authors:**

1. Is there any comparison between the proposed method and the ChatGPT (zero-shot)?
2. Considering the knowledge extraction and fusion modules, does the proposed method  cost more time for training and inference than baselines?
3. Will you make the source code open-sourced? This can benefit the future research that follow your work and improve it.


**Reasons To Accept:**

1. The proposed method is novel and interesting for the community.
2. The proposed model can achieve new state-of-the-art performance.
3. The proposed knowledge integrating modules can fuse the external knowledge adaptively.


**Reasons To Reject:**

1. As ChatGPT is widely adopted for NLP application tasks in nowadays, no results from ChatGPT are provided. If the zero-shot results of ChatGPT can outperform the proposed model, it is hard to highlight the meaning of this work.
2. It is good to include some cases to show the superiority of this work, but there is no error analysis.
3. There are no computation or time cost comparisons with baselines. As mentioned in Introduction, this task is widely applied in real-world scenarios, so computational efficiency should be an important factor.
4. The dataset and source code of this work is not provided. It is hard to reproduce the results reported in this work.

**Reproducibility:**

3: Could reproduce the results with some difficulty. The settings of parameters are underspecified or subjectively determined; the training/evaluation data are not widely available.

**Reviewer Confidence:**

5: Positive that my evaluation is correct. I read the paper very carefully and I am very familiar with related work.

---

> ### Author Rebuttal · Authors · 2023-08-29
>
> Thanks for your careful review and valuable suggestions. We will explain your concerns point by point below.
>
> **Q1: Is there any comparison between the proposed method and the ChatGPT (zero-shot)?**
>
> **A1:** We have evaluated the performance of ChatGPT on the ECTEC task. Specifically, we randomly selected 50 conversations from the test set of ECF, and then fed each test conversation and a task-instruction prompt into ChatGPT to obtain the predicted emotion-cause triplets. The table below presents the zero-shot and few-shot results of ChatGPT. It is obvious that SHARK performs significantly better than ChatGPT in both zero-shot and few-shot settings. We conjecture that due to the complexity of extracting three elements (i.e., emotion, cause, and category) in the ECTEC task, the performance of ChatGPT is not satisfactory.
>
> | Method         | Anger | Disgust | Fear  | Joy   | Sadness | Surprise | 6 Avg. F1 | 4 Avg. F1 |
> | -------------- | ----- | ------- | ----- | ----- | ------- | -------- | --------- | --------- |
> | ChatGPT 0-shot | 13.19 | 21.62   | 28.57 | 24.14 | 14.29   | 16.95    | 18.32     | 17.89     |
> | ChatGPT 5-shot | 14.81 | 20.00   | 0.00  | 33.33 | 26.92   | 15.69    | 22.18     | 22.76     |
> | SHARK (Ours)   | 18.72 | 5.06    | 0.00  | 39.03 | 12.89   | 37.36    | 27.19     | 29.23     |
>
> We will add these results in our revision.
>
> **Q2: There is no error analysis.**
>
> **A2:** Thanks for your nice suggestion! We will add more error analysis in our revision. Based on our observation, the proposed generative framework generally predicts more triplets, and thus significantly improves the *Recall* metric. However, it sacrifices the *Precision* metric to some extent (as shown in the table below). Hence, we plan to analyze several representative error cases regarding imprecise predictions in our revision.
>
> | Method         | 6 Avg. P  | 6 Avg. R  | 6 Avg. F1 | 4 Avg. P  | 4 Avg. R  | 4 Avg. F1 |
> | -------------- | --------- | --------- | --------- | --------- | --------- | --------- |
> | ECPE-2D        | 35.68     | 28.94     | 30.80     | 38.87     | 31.53     | 33.55     |
> | ECPE-2D +  CSK | **36.94** | 26.66     | 30.37     | **40.25** | 29.05     | 33.09     |
> | BART           | 27.70     | **35.19** | 30.35     | 29.23     | **38.14** | 32.74     |
> | SHARK (Ours)   | 30.36     | 35.02     | **32.25** | 31.92     | 37.41     | **34.23** |
>
>
> **Q3: Does the proposed method cost more time for training and inference than baselines?**
>
> **A3:** We have conducted experiments on an Nvidia RTX-3090 GPU to compare the computational efficiency of different methods and report the training and inference time as well as the parameter size.  As shown in the table below, we make the following observations:
>
> - SHARK incurs slightly higher computational cost than the baseline model BART, due to introducing more parameters for the knowledge extraction and fusion modules.
> - Although there are more parameters, the training time and inference time of SHARK are comparable to those of UECA-Prompt.
>
> Based on these observations, we believe that the computational cost of SHARK is generally acceptable considering the performance improvement over other baselines.
>
> | Method             | Number of Parameters (M) | Training Time (s) | Inference Time (s) |
> | ------------------ | ------------------------ | ----------------- | ------------------ |
> | UECA-Prompt        | 108.34                   | 69.19             | 139.52             |
> | UECA-Prompt  + CSK | 108.34                   | 70.31             | 140.67             |
> | BART               | 140.69                   | 55.40             | 118.06             |
> | SHARK (Ours)       | 152.14                   | 77.67             | 144.53             |
>
> *Note*：`Training Time` refers to the duration for the model to run through the training set and the verification set in a single training epoch, and `Inference Time` refers to the time it takes to make predictions on the entire test set. The training/inference time of UECA-Prompt is measured based on its publicly available source codes.
>
> **Q4: Will you make the source code open-sourced?**
>
> **A4:** We will open-source our source codes via GitHub, and add the link in our revision.

---

### Official Review · Reviewer_63gB · 2023-08-05

**Typos Grammar Style And Presentation Improvements:** 1. The writing in the first paragraph…
**Soundness:** 4

**Excitement:**

3: Ambivalent: It has merits (e.g., it reports state-of-the-art results, the idea is nice), but there are key weaknesses (e.g., it describes incremental work), and it can significantly benefit from another round of revision. However, I won't object to accepting it if my co-reviewers champion it.

**Paper Topic And Main Contributions:**

The paper proposes an end-to-end model equipped with commonsense reasoning for the task of emotion-cause triplet extraction. The model utilizes BART to generate triplets, unifying the typically separate subtasks involved in this overall task. This unified approach helps address issues with error propagation that can occur when splitting the task into multiple stages. Additionally, the authors introduce a novel dual-view gate mechanism to incorporate external knowledge into the model, addressing the lack of external knowledge utilization in previous work. Experiments demonstrate the effectiveness of the proposed model on this challenging task.

**Reasons To Accept:**

1. The paper's writing is solid, with the methods section being particularly well-structured and easy to follow.
2. The proposed generative-based model is the first unified model for the emotion-cause-triplet extraction task.
3. The authors conducted extensive experiments, including re-implementing other baselines, and the achieved performance scores are impressive, demonstrating the model's effectiveness.

**Reasons To Reject:**

1. The generative-based model proposed in the paper may not be considered novel, as similar approaches exist in general information extraction or sentiment analysis areas. The issue of error propagation is also common in these tasks, and it is not specified for emotion-cause-triplet extraction. To strengthen the motivation behind the proposed model, the authors could provide a more persuasive argument that highlights the unique challenges of emotion-cause-triplet extraction and how the generative model is well-suited to handle them. The core of this task revolves around understanding how one utterance causes an emotion in another utterance, and further clarification on this aspect would be beneficial.
2. In the experiment part, the fine-grained analysis regarding the function of the specific module is missing, which results in the necessity of the proposed module not being fully verified. The overall score and ablation study show the results from a high-level view. However, there should be more fine-grained, numerical analysis regarding the generative-based model and CKS. While a case study can offer an intuitive understanding of the module’s performance, it is essential to ensure that the results are not biased by carefully selected cases. Therefore, it would be beneficial to examine what kinds of cases or problems the module, e.g., generative module, is oriented to solve and to what degree the module improves the performance for these specific cases.

**Reproducibility:**

3: Could reproduce the results with some difficulty. The settings of parameters are underspecified or subjectively determined; the training/evaluation data are not widely available.

**Reviewer Confidence:**

4: Quite sure. I tried to check the important points carefully. It's unlikely, though conceivable, that I missed something that should affect my ratings.

---

> ### Author Rebuttal · Authors · 2023-08-29
>
> Thanks for your careful review and valuable suggestions. We will explain your concerns point by point below.
>
> **Q1: About the motivation for solving the ECTE task with the generative model.**
>
> **A1:** We clarify our motivation as follows:
>
> To address the ECTEC task, an intuitive idea is to first identify which utterance expresses the emotion, and then find the cause utterance that triggers it, followed by determining the emotion category based on the emotion-cause utterance pair. Therefore, emotion is the core element of the ECTEC task, and the other two elements (i.e., cause and category) are highly dependent on the emotion utterance. We believe the decoder in the generative model can well capture the conditional dependency relationship among the three elements, because it generates the emotion utterance, cause utterance, and emotion category in an autoregressive manner to form the triplet.
>
> Thanks for your constructive comments. We will add this argument in our revision to strengthen the motivation behind the proposed generative model.
>
> **Q2: The fine-grained analysis regarding the function of the specific module is missing.**
>
> **A2:** We report the fine-grained experimental results on the ECF dataset in the table below. In general, compared with previous encoder-based methods (ECPE-2D), the generative module (BART) significantly boosts the *Recall* metric, i.e., 6.25% and 6.61% in two weighted average scores, respectively. In addition, comparing the results of BART and SHARK, it is easy to find that the integration of CSK (SHARK) brings a significant increase in the *Precision* metric, i.e., 2.66% and 2.69% in two weighted average scores, respectively.
>
> | Method         | 6 Avg. P  | 6 Avg. R  | 6 Avg. F1 | 4 Avg. P  | 4 Avg. R  | 4 Avg. F1 |
> | -------------- | --------- | --------- | --------- | --------- | --------- | --------- |
> | ECPE-2D        | 35.68     | 28.94     | 30.80     | 38.87     | 31.53     | 33.55     |
> | ECPE-2D +  CSK | **36.94** | 26.66     | 30.37     | **40.25** | 29.05     | 33.09     |
> | BART           | 27.70     | **35.19** | 30.35     | 29.23     | **38.14** | 32.74     |
> | SHARK (Ours)   | 30.36     | 35.02     | **32.25** | 31.92     | 37.41     | **34.23** |
>
> Moreover, we conduct further analysis to explore the impact of different conversation lengths, i.e., separately evaluating the predictions for conversations with varying numbers of utterances in the test set. As shown in the following table, we can observe that our generative models perform much better than encoder-based methods on conversations with more than 10 utterances. In these long conversations, encoder-based methods often fail to fully consider the contextual information and ignore a number of triplets, while our generative models tend to comprehensively capture the context, effectively capturing a broader range of triplets.
>
> | num_utt ≤ 10   |              |              |               |              |              |               |
> | -------------- | ------------ | ------------ | ------------- | ------------ | ------------ | ------------- |
> | **Method**     | **6 Avg. P** | **6 Avg. R** | **6 Avg. F1** | **4 Avg. P** | **4 Avg. R** | **4 Avg. F1** |
> | ECPE-2D        | 35.57        | 33.22        | 33.56         | 39.65        | 37.04        | 37.41         |
> | ECPE-2D +  CSK | 35.94        | 30.73        | 32.84         | 40.07        | 34.26        | 36.61         |
> | BART           | 28.69        | 36.54        | 31.45         | 30.56        | 40.37        | 34.50         |
> | SHARK (Ours)   | 31.27        | 37.18        | 33.68         | 33.46        | 40.00        | 36.16         |
>
> | num_utt > 10   |              |              |               |              |              |               |
> | -------------- | ------------ | ------------ | ------------- | ------------ | ------------ | ------------- |
> | **Method**     | **6 Avg. P** | **6 Avg. R** | **6 Avg. F1** | **4 Avg. P** | **4 Avg. R** | **4 Avg. F1** |
> | ECPE-2D        | 36.00        | 26.91        | 29.41         | 38.81        | 29.01        | 31.71         |
> | ECPE-2D +  CSK | 38.02        | 24.73        | 29.10         | 40.99        | 26.66        | 31.37         |
> | BART           | 27.60        | 34.56        | 29.91         | 28.89        | 37.12        | 32.03         |
> | SHARK (Ours)   | 29.77        | 34.26        | 31.49         | 31.50        | 36.47        | 33.46         |
>
>
> **Q3: The writing in the first paragraph lacks smoothness. I'm unsure why the author opted to begin with a discussion on 'social media'.**
>
> **A3:** Thanks for pointing out this. We will adopt your nice suggestion to rewrite the first paragraph in our revision.

---

### Meta-Review · Area_Chair_bVwv · 2023-09-19

**Recommendation:** 3

**Metareview:**

The paper proposes an end-to-end model for emotion-cause triplet extraction which is augmented with commonsense knowledge. The unification proposed in this paper aims to better handle error propagation. The generated and retrieved commonsense knowledge are integrated into the model with a novel gating mechanism. The results exhibit the effectiveness of the proposed model. The paper was missing some key details which the rebuttal helped addressing. The additional inference/training time footprint argument and the argument about its closeness to the existing approaches (i.e., UECA-Prompt) is rather unconvincing when we consider those numbers at large scale. The writing, in particular the introduction, needs further improvement to better position the problem.

---

### Decision · Program_Chairs · 2023-10-07

**Decision:**

Accept-Findings

**Comment:**

The paper proposes an end-to-end model for emotion-cause triplet extraction which is augmented with commonsense knowledge. The unification proposed in this paper aims to better handle error propagation. The generated and retrieved commonsense knowledge are integrated into the model with a novel gating mechanism. The results exhibit the effectiveness of the proposed model. The paper was missing some key details which the rebuttal helped addressing. The additional inference/training time footprint argument and the argument about its closeness to the existing approaches (i.e., UECA-Prompt) is rather unconvincing when we consider those numbers at large scale. The writing, in particular the introduction, needs further improvement to better position the problem.